# NON-SEQUENTIAL MELODY GENERATION

## ABSTRACT

In this paper we present a method for algorithmic melody generation using a generative adversarial network without recurrent components. Music generation has been successfully done using recurrent neural networks, where the model learns sequence information that can help create authentic sounding melodies. Here, we use DCGAN architecture with dilated convolutions and towers to capture sequential information as spatial image information, and learn long-range dependencies in fixed-length melody forms such as Irish traditional reel.

## 1 INTRODUCTION

Algorithmic music composition is almost as old as computers themselves, dating back to the 1957 "Illiac suite" (Hiller Jr & Isaacson, 1958). Since then, automated music composition evolved with technology, progressing from the first rule-and-randomness based methods to the sophisticated tools made possible by modern-day machine learning (see Fernández & Vico (2013) and Briot et al. (2017) for detailed surveys on history and state of the art of algorithmic music composition).

One of the first machine learning (ML) approaches to music generation was Conklin & Witten (1995), who used the common notion of entropy as a measurement to build what they termed a multiple viewpoint system. Standard feedforward neural networks have difficulties with sequence based information such as music. Predicting the next note of a piece, when only based on the current note, does not account for long-range context or structure (such as key and musical sections) which help give coherence to compositions. As music is traditionally represented as sequences of notes, recurrent neural networks are a natural tool for music (especially melody) generation, and multiple groups used RNNs fairly successfully for a variety of types of music. Todd (1989) used a sequential model for composition in 1989, and Eck & Schmidhuber (2002) used the adapted LSTM structure to successfully generate music that had both short-term musical structure and contained the higher-level context and structure needed. Subsequently, there have been a number of RNN-based melody generators (Simon & Oore, 2017; Lee et al., 2017; Eck & Lapalme, 2008; Sturm et al., 2016; Hadjeres et al., 2017; Chen & Miikkulainen, 2001; Boulanger-Lewandowski et al., 2012; Yu et al., 2017). Other approaches such as MidiNet by Yang et al. (2017), though not RNNs, also leveraged the sequential representation of music.

Using an RNN architecture provides a lot of flexibility when generating music, as an RNN has the ability to generate pieces of varying length. However, in some styles of music this is not as desired. This is true of traditional Irish music - and especially their jigs and reels. These pieces have a more rigid format where the varying length can prevent capturing the interplay between the phrases of the piece.Finding jigs and reels to train on was made easy by an excellent database of Irish traditional melodies in ABC notation (a text based format), publicly available at TheSessionKeith. Several RNN-based generators were trained on the melodies from TheSession, most notably Sturm et al. (Sturm et al., 2016; Sturm & Ben-Tal, 2018), as well as Eck & Lapalme (2008).

It is natural to view music, and in particular melodies, as sequential data. However, to better represent long-term dependencies it can be useful to present music as a two-dimensional form, where related parts and occurrences of long patterns end up aligned. This benefit is especially apparent in forms of music where a piece consists of a well-defined, fixed-length components, such as reels in Irish music. These components are often variations on the same theme, with specific rules on where repeats vs. changes should be introduced. Aligning them allows us to use vertical spatial proximity to capture these dependencies, while still representing the local structure in the sequence by horizontal proximity.

In this project, we leverage such two-dimensional representation of melodies for non-sequential melody generation. We focus on melody generation using deep convolutional generative adversarial networks (DCGANs) without recurrent components for fixed-format music such as reels. This approach is intended to capture higher-level structures in the pieces (like sections), and better mimic interplay between smaller parts (musical motifs). More specifically, we use dilations of several semantically meaningful lengths (a bar or a phrase) to further capture the dependencies.

Dilated convolutions, introduced by Yu & Koltun (2015), have been used in a number of applications over the last several years to capture long-range dependencies, notably in WaveNet (Oord et al., 2016). However, they are usually combined with some recurrent component even when used for a GAN-based generation such as in Zhang et al. (2019) or Liu & Yang (2019).

Not all techniques applicable to images can be used for music, however: pooling isn't effective, as the average of two pitches can create notes which fall outside of the 12-semitone scale (which is the basis the major and minor scale as well as various modes). This is reflected in the architecture of our discriminator, with dilations and towers as the main ingredients.

## 2 STRUCTURE OF THE DATA: TRADITIONAL IRISH TUNES

In spite of emigration and a well-developed connection to music influences from its neighbouring cultures - especially Scotland - Irish traditional music has kept many of its elements and itself influenced many forms of music. Its influences can be found in the folk music of Newfoundland and Quebec and in Bluegrass from the Appalachian region of United States.

A significant part of the traditional Irish music is instrumental dance music, where a tune following a set format is played on a solo instrument, such as a fiddle, tin whistle, accordion or flute. The core of the piece is usually monophonic, so while harmonies and embellishments can be introduced during performances, they are considered extra and not usually encoded when a piece is transcribed. Generally the melody consists of two distinct 8-bar parts, which in turn consist of two 4-bar phrases each. Usually, phrases represent a pattern which repeats with variations from phrase to phrase, with one (most often third) phrase deviating most from the pattern, and the final phrase returning to it. When performing, each part is repeated twice; sometimes the second repeat of a part is different enough that it is transcribed separately, making the transcription from 16 to 32 bars long. The whole piece is usually performed three (or more) times, especially when accompanying a dance.

The two most common types of dances are reels and jigs: the reels have time signature of 4/4 (that is, each bar contains 4 units, where each unit is a quarter note), whereas a jig has a time signature 6/8 (that is, each bar contains 6 units, where each unit is an eighth note).

Many Irish tunes are in a major key, with D major being most common. However, a number of tunes use the Dorian and Mixolydian modes (starting with 2nd or 5th note with respect to the relative major scale, respectively). The natural minor key (Aeolian mode) is less common. Overall, it is uncommon for a tune to incorporate notes not in its key and to go outside of a 2-octave range (either of these would make it hard to play on a tin whistle).

Traditionally, musicians learned tunes by ear. However, the relative simplicity of the structure of Irish tunes has lent itself well to a text-based representation, culminating in the ABC notation developed by Chris Walshaw in the 1970s. There, the key and default length of a note are specified in a header (together with the title, time signature, etc), and the melody is then encoded by letters: A-G ,where C represents middle C, and a-g, which represents the notes an octave higher than A-G. Additionally there are modifiers which raise and lower notes by an octave, as well as charaters which represent sharps and flats. Like in the staff notation, sharps and flats which are implied by the key are not explicitly written into the piece. These letters are also appended with duration modifiers: for example, if ABC header sets the default note length as 1/8, then D represents an eighth note (which is a D), D2 represents a quarter note, and D/2 represents a sixteenth note. For readability, bar separators | and spacing are often included: for example, the sequence "d2dA BAFA|ABdA BAFA|ABde f2ed|Beed egfe" encodes the first phrase of a reel called "Merry Blacksmith", where the key was specified to be D major and the default note length is set to 1/8.

Nowadays, ABC notation is widely used, especially for Irish folk music, with additional features for representing ornaments, repetitions, and so on. In particular, this is the default representation for tunes in databases such as TheSession which we used for this project.

## 2.1 Preprocessing TheSession dataset

TheSession dataset contains over 31,000 tunes, which contain 10,000 reels in a major key. However, after removing improperly formatted tunes, tunes with features we did not want to generate (such as triplets), and tunes that had more than 16 bars we were left with 820 reels, which we used as our training data.

The samples in ABC notation were then converted into numerical vectors, with 16 numbers per bar (so 256 numbers total), corresponding to 16 possible notes (normalized midi values). This encoding did not preserve information about the duration of each note, only their (normalized) midi value. However, we found that with simple postprocessing (converting a sequence of occurrences of the same note into a note of a longer duration, starting with an occurrence on a beat) we were able to recover the tunes with not much more variation than would be introduced in performances.

One of the main ideas behind our encoding was representing the resulting vectors of 256 values as a 64x4 image, with midi values corresponding to pixel values. The resulting image had a line for each the phrase of the tune, and the vertical alignment of the corresponding notes and bars let us exploit the long-range dependencies among the phrases using their spatial proximity. Intuitively, this is akin to building a mental map of a space explored by touch: though the touch information is sequential, the resulting map can be two or three-dimensional, with significant amount of information encoded by proximity in the second and third dimensions.

## 3 GAN Architecture

Traditionally, sequential data is learned using some variation of RNNs. Simple implementations of RNNs (standalone GRU and LSTM cells) tend to struggle with capturing long-distance dependencies; a popular solution to this issue is to use bi-directional RNNs (that is, a pair of layers that run in opposite directions) with an activation mechanism (using an auxiliary trainable network to better relate elements of the output sequence to those of the input sequence) so that information for multiple sections of the sequence can be captured(Zhou et al., 2016).

A similar effort at capturing long-distance dependencies has been applied to Convolutional Neural Networks (CNNs) too. Yu et al. have proposed *dilated convolutions* as a solution to the problem: in a dilated convolution, the convolution kernel is not a contiguous block, but rather subsections of the kernel are separated by some number of pixels from each other. As a result, the convolution operation is not limited to the immediate neighbourhood, and long-distance dependencies are made available to the kernel (Yu & Koltun, 2015). While attention-based and bi-directional RNNs have proven to be quite successful at capturing long-distance dependencies, we believe that there are a few notable benefits for being able to use a DCGAN for generating sequential data, as opposed to an RNN:

- DCGANs that incorporate dilations allow for domain-specific heuristics to be passed to the neural network, by means of the dilation rates and the convolution kernel dimensions. This allows for a whole new dimension of flexibility and fine-tuning that is not available through RNNs.

- A DCGAN yields *both* a discriminator and a generator, as opposed to just a generator.

- Unlike an RNN, a GAN does not require a meaningful or domain-specific seed to be passed as an input to the generator; a vector of random noise can be used instead, making it easier to generate novel outputs from the generator.

GANs have traditionally worked well with pictures, but to our knowledge existing music generation with GANs either relies on RNNs in the discriminator, as in SeqGAN (Yu et al., 2017) or generates the melody in a sequential bar-after-bar process as in MidiNet (Yang et al., 2017).

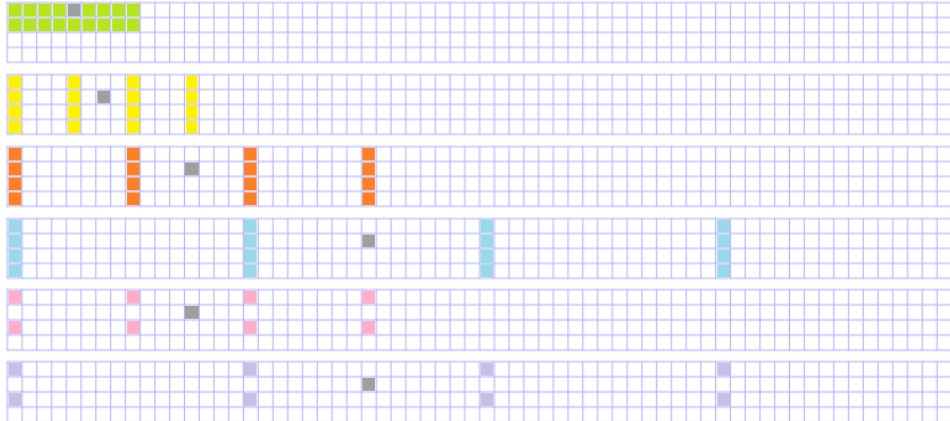

Figure 1: Highlighted pixels show convolution patterns for each tower, with the grey pixel being the center of each kernel.

## 3.1 MODEL

### 3.1.1 DISCRIMINATOR

The discriminator starts off with a 6-tower convolution network (that is, 6 convolutions run side-by-side, as opposed to being dependent on each other). We chose to use dilated filters on 5 of the towers to capture relative information between bars and phrases (we shall refer to this as the *global context*), with the remaining layer being a contiguous 2x9 convolution to also capture the structure of the immediate neighbourhood (*local context*). Having multiple towers each learning different aspects gives us versatility and helps avoid the "tunnel vision" nature of other generative music models. This can be viewed in Figure 1.

All the towers use 32 convolution filters, and use zero-padding for edge pixels that might not have a sufficient neighbourhood to be covered by each filter. The outputs of the convolutions from each tower are then stacked horizontally, and passed to a second (regular) layer of convolution made of 64 filters, using a 3x3 kernel, a 2x2 stride and with the convolution being only applied to pixels that have a sufficient neighbourhood for the convolution filter. The output of this layer is then flattened, and passed to a dense layer consisting of 1024 neurons, which is in turn followed by a sigmoid neuron that generates the prediction of the discriminator. No batch normalization is applied to any of the layers on the discriminator.

### 3.1.2 GENERATOR

Unlike the discriminator, the generator does not make use of dilations or towers. We start off with a dense layer made of 32x2x256 (where the last dimension is the number of filters). The dense layer is then reshaped into 256 filters of size 32x2, and passed through two layers of deconvolution, each with halving the number of filters, while using a stride of 1x1 and a kernel size of 2x5. A last deconvolution layer (also 2x5 in kernel size) is used to bring down the filters into one image, with a stride of 2x2 to return to the original image dimension of 64x4; this layer also uses a tanh activation to place the output values in the same range as the training dataset. All layers in the generator use batch normalization.

## 3.2 POST-PROCESSING

After our GAN has generated music in the form of a 64x4 matrix, we map the real values to notes in ABC notation, rounding to the nearest note in the D major key. We then take each beat and merge notes with the same pitch together, so a sequence of four 16th notes C starting from a 1st, 5th, 9th or 13th note in a bar becomes a quarter note C instead. The resulting ABC notated music can then be converted into sheet music or played with midi generators.

## 4 COMPARISON WITH OTHER MELODY GENERATORS

Our approach is similar to the model by Sturm et al. (2016) as both train on text-based musical notation and aim to learn Irish traditional instrumental music that has distinct structure. The models in Magenta are considered to be state of the art in music generation at the moment. For this reason, we compared our samples with the Folk-RNN and Magenta models - specifically the MelodyRNN model in Magenta - training using the same dataset of 820 curated reels. We generated an equivalent number of samples from each model and compared the samples. Since our goal is for the GAN to learn the global patterns, we use measures that highlight similarity in structure.

The first metric we chose was the normalized mean Fréchet distance. Fréchet distance is a measure of similarity between vectors which prioritizes the ordering and magnitude of the values in those vectors. Since the order and distribution of notes are important for a tune to be considered structurally similar to another, this metric is ideal for us. We normalize this Fréchet distance so we can view the changes between the phrases better.

The results of this comparison are found in Figure 2. Each group records the normalized Fréchet distance between corresponding combinations of phrases i.e the first group highlights the normalized mean Fréchet distance between phrase 1 and 2 for all the distributions. We can see that our generated samples compare favorably with the Folk-RNN model and both exhibit similar structure to the training set. It is important to note that the MelodyRNN used here is the default model without the attention mechanism.

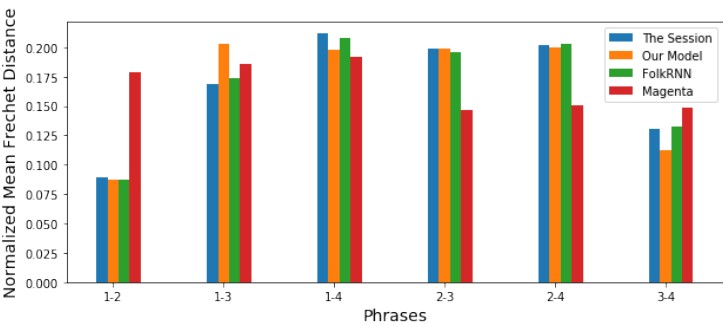

Figure 2: Average normalized Fréchet distance between the phrases of the distributions

Another measure to visualize similarity between distributions is the t-distributed stochastic neighbourhood embedding (t-SNE) algorithm developed byMaaten & Hinton (2008). This algorithm groups points that are similar in higher dimension closer together in low dimensions - making it useful to visualize similarity between the tunes generated by both models along with the training data. Here, we set the tunable parameter perplexity to 50. This increases the importance of global similarity between vectors.

Figure 3 shows distributions of the training data and samples generated by our model, FolkRNN and Magenta trained on the same data. Since t-SNE is stochastic and can generate different visualizations for each run, we display visualization produced by two runs of t-SNE algorithm. It can be observed that most of the time (>75%), the tunes generated by our model seem to lie within the distribution of the training data, although they only cover a subset of the training data. Both RNN models seem to mimic a slightly different distribution, where we can think of the tunes generated by Magenta as a subset of the FolkRNN ones, with FolkRNN generating by far the most diverse set of samples.

A third metric is looking at the distribution of notes in the samples. Figure 4 shows the frequency of distribution of notes as midi key values in the samples. As music theory would suggest, the tonic of the key (in this case, the midi value corresponding to D) is the most common, with the 3rd and 5th of the key more frequent than the rest of the scale. Both our model and FolkRNN created a set of samples which mimics this distribution of the notes in the key, however Magenta seems to have a more uniform distribution of note frequencies.

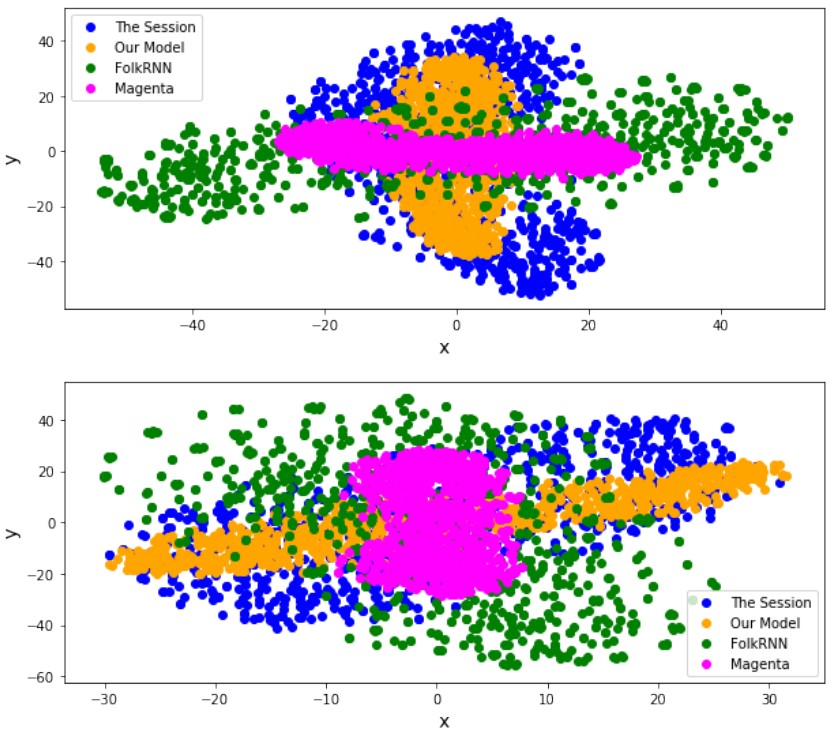

Figure 3: t-SNE visualization of the distributions of tunes

Overall, these metrics show that music generated by our DCGAN is comparable to melodies generated by RNNs on the same symbolic data. The use of dilations allows for our model to learn global tune structure.

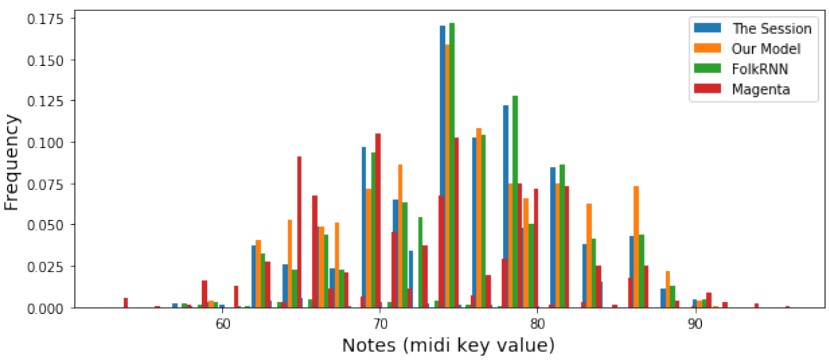

Figure 4: Distribution of notes

# 5 CONCLUSION

Converting sequential data into a format which implicitly encodes temporal information as spatial information is an effective way of generating samples of such data as whole pieces. Here, we explored this approach for melody generation of fixed-length music forms, such as an Irish reel, using non-recurrent architecture for the discriminator CNN with towers and dilations, as well as a CNN for the GAN itself.

One advantage of this approach is that the model learns global and contextual information simultaneously, even with a small model. LSTMs and other approaches need a much larger model to be able to learn both the contextual neighboring note sequences and global melody structure.

In future work, we would like to introduce boosting in order to capture the structure of the distribution more faithfully, and increase the range of pieces our model can generate. Natural extensions of the model would be to introduce multiple channels to capture durations better (for example, as in Colombo et al. (2016)), and add polyphony (ie, using some form of piano roll representation). Another direction could be to experiment with higher-dimensional representation of the sequence data, to better capture several types of dependencies simultaneously. Additionally, it would be interesting to apply it to other kinds of fixed-length sequential data with long-range patterns.

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
