# OpenReview forum: "Non-Sequential Melody Generation"
_ICLR.cc/2020/Conference — Reject_

### Official Review · AnonReviewer3 · 2019-10-18
**Official Blind Review #3**

**Rating:** 1

**Review:**

This paper presents a DCGAN for generating Irish folk music. The fixed length of Irish reels (in terms of bars) means that a piece can be represented as a 2-D image with fixed height and fixed width. This study represents each performance as a 4x64 image, where each successive row represents a phrase and pixels in neighbouring rows represent corresponding notes in successive phrases. A DCGAN architecture is then used to generate novel melodies using this representation. Each pixel has a normalised value between [-1,1] which correspond to the notes A-G, a-g in the vocabulary.

This study to me appears incomplete and at an early stage and requires further work before it can be accepted as a conference publication. Firstly, I believe the title is misleading. The title encourages the reader to expect a general model for generating melodies with GANs however the application is specifically to generate monophonic Irish folk music only, with a fixed number of bars. Secondly, the main novelty in the paper, the DCGAN architecture has not been described in enough detail, for instance the optimiser and hyper parameters are not mentioned, making it difficult for this work to be reproduced.

The authors claim 3 major motivations for  the DCGAN; 1) Dilated convolutions for introducing music related inductive priors, however the model description does not provide any intuition or insight for why the exact filters used were chosen. 2) The authors claim the GAN also yields a discriminator in addition to a melody generator, but they do not provide any explanation for why a discriminator might be useful. 3) They claim using random noise instead of a musically meaningful seed is better, but they do not describe the input noise distribution that the proposed generator is conditioned on.

For the evaluations, the Frechet distance is introduced without a reference/citation and the “models in Magenta” are referenced without a citation.  One downside of using the GAN is that there is no way to calculate the log-likelihood of the generated samples under the model distribution, which makes evaluating/comparing models a difficult problem. I think the distribution of notes in the generated samples is not a very informative metric for comparing the quality of generated samples. A few audio examples would have been extremely useful in getting a sense of how this model performs.

I think this work should be resubmitted with major revisions.

Minor Comments

1. Section 2 does a good job of providing background for Irish folk music and the ABC notation.
2. “normalised MIDI value” should be elaborated. I inferred it to mean the MIDI values are normalised to the range [-1,1] after reading the whole paper, however this is not clear in Section 2.1
3. “resulting vectors of 256 values as a 64x4 image”, I think the image is 4x64 (Figure 1).
4. “..to use bidirectional RNNs with an activation mechanism”, this should be attention mechanism.
5. There should be some comment on the motivation behind the specific choice of filters in the generator and discriminator.
6. There should be some details about the optimisation algorithm, batch sizes, learning rate schedules etc used to train the network.
7. The magenta model requires a citation as it is not obvious what is being referred to.
8. The Frechet distance should be introduced with a citation.





**Experience Assessment:**

I have published in this field for several years.

**Review Assessment: Checking Correctness Of Derivations And Theory:**

N/A

**Review Assessment: Checking Correctness Of Experiments:**

I assessed the sensibility of the experiments.

**Review Assessment: Thoroughness In Paper Reading:**

I read the paper thoroughly.

---

> ### Author Response · Authors · 2019-11-15
> **Thank you**
>
> Thank you for your feedback. We appreciate the meticulousness you put into reading the paper, and will use add those changes in the revision.
>
> We were also wondering if you had any ideas of what metrics might be better than the note distribution? We opted to use this, as in the beginning of the project, the generated music very obviously fell outside what would be typically expected, and this metric helped quickly asses large amounts of data without listening to many songs.

---

### Official Review · AnonReviewer1 · 2019-10-22
**Official Blind Review #1**

**Rating:** 3

**Review:**

This paper describes a (DC)GAN architecture for modeling folk song melodies (Irish reels).
The main idea of this work is to exploit the rigid form of this type of music --- bar structure and repetition --- to enable 2-dimensional convolutional modeling rather than the purely sequential modeling that is commonly used in recent (melodic) music generation architectures.
The ideas in this paper seem sound, though they primarily consist of recombining known techniques for a specific application.
The main weaknesses of this paper are in the evaluation (see below), and while I understand that evaluating generate models for creative applications is difficult and fraught territory, I don't think the efforts taken here are sufficiently convincing.



Strengths:

- The paper is clearly written, and the authors have taken great care to describe the unique structures of the data they are modeling.
- The proposed architecture seems well motivated, and matches to the structure of the data.


Weaknesses:

There are three components to the evaluation, and each of them are problematic:

- The first evaluation (Fig 2) compares the average Frechet distance between phrases generated by different models, and within the original dataset.  Some brief argument is given for why Frechet is a good choice here, but it still seems quite tenuous: what does this distance intuitively mean in terms of the data?  How should the scale of these distances be interpreted / what's a meaningfully large difference?  How concentrated are these average distances (ie, please show error bars, variance estimates, or some notion of spread)?

- The second evaluation (Fig 3) uses t-SNE to embed the generated melodies into a 2D space to allow visual inspection of the differences between distributions produced by each model.  While this might be a reasonable qualitative gut-check, t-SNE is by no means an appropriate tool for quantitative evaluation.  The authors at least did multiple runs of t-SNE, but this hardly amounts to compelling evidence.  Moreover, combining all data sources into one sample prior to running t-SNE induces dependencies between the point-wise neighbor selection distributions, which seems undesirable if the eventual goal is to determine how similar each model's distribution is to the source data.  A better approach might be to create independent plots for each model's output (with the original data), but I'd generally advise against using t-SNE for this kind of analysis altogether.

- The third evaluation (Fig 4) measures the amount of divergence from the key (D) in terms of note unigrams.  This evaluation is done qualitatively, and the histogram is difficult to read --- it may be easier to read if the octave content was collapsed out to produce pitch classes rather than pitches.  If, however, the goal is to actually measure distance from the target key, one could do this quantitatively by comparing histograms to a probe tone profile (or otherwise constructed unigram note model) to more clearly characterise the behaviors of the various models in question.


At a higher level, there is no error analysis provided for the model, nor any ablation study to measure the impact of the various design choices taken here (eg dilation patterns in Figure 1).
The authors seem to argue that these choices are the main contribution of this work, so they should be explicitly evaluated in a controlled setting.


**Experience Assessment:**

I have read many papers in this area.

**Review Assessment: Checking Correctness Of Derivations And Theory:**

I carefully checked the derivations and theory.

**Review Assessment: Checking Correctness Of Experiments:**

I assessed the sensibility of the experiments.

**Review Assessment: Thoroughness In Paper Reading:**

I read the paper thoroughly.

---

> ### Author Response · Authors · 2019-11-15
> **Thank you**
>
> Thank you for your feedback. We particularly like the suggestion of comparing the different iterations of the models we tried.
>
> As well, when it comes to the T-SNE metric we used, do you have something specific in mind that you feel would be better than it?

---

### Official Review · AnonReviewer2 · 2019-10-26
**Official Blind Review #2**

**Rating:** 1

**Review:**

The paper is interested in music generation, leveraging a 2D representation of the music data.

The idea of using a 2D representation is quite good, though here very specific of the considered type of music. The authors might want to discuss related approaches , considering 2D general representations of music [1, 2]. The originality of the proposed approach is to use dilated convolutions to capture the long range dependencies in a GAN framework.

The paper however looks premature for publication at ICLR, for several reasons:

* The thorough discussion of the dataset might be put in supplementary material. Instead, the reader would like to know whether the authors considered data augmentation of their relatively small dataset (820 tunes). Likewise, the description of the architecture could be put in a supplementary section, or on github for reproducibility.

* The authors did not justify the choice of the specifics of the architecture, such as the number of filters, layers or the presence / absence of batch normalization (part 3.1). The reviewer would like to see how the results vary with respect to those parameters. It would also be interesting to see what the generated images look like before the tanh (part 3.1.2).

* The assessment of the results is hard to interpret; the reasons why the Frechet distance varies depending on the models and the phrases combinations should be discussed.

* The gold standard for evaluating music is based on the human assessment of the generated music (involving naive, advanced and expert people), as done in [3], cited; having a human being evaluate and compare the tunes would help to gain insight into the generation. In the same perspective, it would be appreciated to put the results on a website for the reader to assess the quality of the generated music.

* The contraction of the support (as displayed in Fig. 3) suggests that there might be some mode dropping with the GAN; this should be studied in depth.

* The authors consider Magenta and FolkRNN as baselines; the reviewer suggests to also consider e.g. [4]  (although not not exploiting the specifics of the style), to comparatively assess the proposed approach.

[1] "Onsets and Frames: Dual-Objective Piano Transcription", Hawthorne et al., 2017.
[2] "TimbreTron: A WaveNet(CycleGAN(CQT(Audio))) Pipeline for Musical Timbre Transfer", Huang et al., 2018.
[3] "DeepBach: a Steerable Model for Bach Chorales Generation", Hadjeres et al., 2017
[4] "Music Transformer", Huang et al., 2018.

**Experience Assessment:**

I have read many papers in this area.

**Review Assessment: Checking Correctness Of Derivations And Theory:**

N/A

**Review Assessment: Checking Correctness Of Experiments:**

N/A

**Review Assessment: Thoroughness In Paper Reading:**

I read the paper at least twice and used my best judgement in assessing the paper.

---

> ### Author Response · Authors · 2019-11-15
> **Thank you for your feedback**
>
> Thank you very much for your thorough feedback and pointers to many useful references.  We will incorporate your suggestions into the next version of the paper, in particular including a more detailed explanation and evaluation of hyperparameters, more comparisons, and a further evaluation with respect to both given and other metrics.  We are especially grateful for your suggestions on the analysis of mode dropping, data augmentation and variants of the architecture to consider.

---

### Decision · Program_Chairs · 2019-12-19

**Decision:**

Reject

**Comment:**

All the reviewers pointed out issues with the experiments, which the rebuttal did not address. The paper seems interesting, and the authors are encouraged to improve it.